# A root cap-localized NAC transcription factor controls root halotropic response to salt stress in Arabidopsis

Lulu Zheng [1,2,3], Yongfeng Hu[4], Tianzhao Yang[5], Zhen Wang[1], Daoyuan Wang[1], Letian Jia[1], Yuanming Xie [1], Long Luo[5], Weicong Qi[6], Yuanda Lv [6], Tom Beeckman [2,3], Wei Xuan [1] ✉ & Yi Han [5] ✉

Plants are capable of altering root growth direction to curtail exposure to a saline environment (termed halotropism). The root cap that surrounds root tip meristematic stem cells plays crucial roles in perceiving and responding to environmental stimuli. However, how the root cap mediates root halotropism remains undetermined. Here, we identified a root cap-localized NAC transcription factor, SOMBRERO (SMB), that is required for root halotropism. Its effect on root halotropism is attributable to the establishment of asymmetric auxin distribution in the lateral root cap (LRC) rather than to the alteration of cellular sodium equilibrium or amyloplast statoliths. Furthermore, SMB is essential for basal expression of the auxin influx carrier gene *AUX1* in LRC and for auxin redistribution in a spatiotemporally-regulated manner, thereby leading to directional bending of roots away from higher salinity. Our findings uncover an SMB-AUX1-auxin module linking the role of the root cap to the activation of root halotropism.

Plant roots exhibit plasticity to sense and cope with ever-changing environmental cues in soil, including water and nutrient status, and abiotic stresses. A compelling way in which plants can adapt to their environment is by modifying their root system architecture[1]. Plant roots can also adjust their root growth direction towards or away from specific stimuli; this process is known as a root tropic response and likely contributes to plant acclimatory responses to environmental stresses[2–5]. Increased soil salinity is a notable challenge in agriculture, as salt impedes plant growth and reduces crop yield[6]. Approximately 6% of the world's total land area is threatened by excess salinity[7], and this problem continues to worsen. Thus, unraveling the mechanism of plant response to salt and enhancing plant salt tolerance are fundamental to agricultural production. Plant root system architecture is indispensable for plant survival and productivity in response to salinity

stress. Salt stress can activate positive (alternative) pathways to contribute to root development modulation, while it has global negative effects on growth regulatory pathways[8,9]. Furthermore, plants can activate directional root bending away from the high saline conditions to avoid salt stress, termed halotropism[10]. This negative tropism is specific to excessive sodium ($Na^+$) in a dose-dependent manner rather than to secondary osmotic stress[10]. The Cholodny–Went theory is applicable to such a salt avoidance response. Consistent with this well-illustrated theory, the phytohormone auxin is translocated from the stressed to the non-stressed side of the root tip, leading to asymmetric auxin distribution and root bending[10].This auxin movement occurs at the root cap and epidermal cells and is facilitated by auxin transporters[9]. Salt stress-triggered internalization of the auxin efflux carrier PIN-FORMED 2 (PIN2), on the salt-exposed side of the roots,

[1]Sanya Institute of Nanjing Agricultural University, State Key Laboratory of Crop Genetics & Germplasm Enhancement, College of Resources and Environmental Sciences, Nanjing Agricultural University, Nanjing, China. [2]Department of Plant Biotechnology and Bioinformatics, Ghent University, Technologiepark 71, Ghent B-9052, Belgium. [3]VIB-UGent Center for Plant Systems Biology, Technologiepark 71, Ghent B-9052, Belgium. [4]Key Laboratory of Three Gorges Regional Plant Genetics and Germplasm Enhancement, Biotechnology Research Center, China Three Gorges University, Yichang, China. [5]National Engineering Laboratory of Crop Stress Resistence Breeding, School of Life Sciences, Anhui Agricultural University, Hefei 230036, China. [6]Excellence and Innovation Center, Jiangsu Academy of Agricultural Sciences, Nanjing 210014, China. ✉e-mail: wexua@njau.edu.cn; yi.han@ahau.edu.cn

allows rapid asymmetric auxin distribution. Phospholipases, including PLDζ1 and PLDζ2, are mainly responsible for PIN2 internalization[11]. Accordingly, on the non-salt-exposed side of the roots, increased protein abundance of the auxin influx carrier AUX1 co-facilitates rapid changes in auxin flow[12]. However, much less is known about exactly how this auxin influx is regulated in response to halo-stimulation.

The root cap is located at the most distal end of the root tip and includes the central columella, and the peripheral lateral root cap (LRC)[13]. Apart from safeguarding the apical meristem within the root tip, the root cap functions in sensing environmental cues and transducing these signals to optimize the growth of the root[13], including periodic lateral root formation[14,15], gravitropism[16,17], phototropism[18], and thigmotropism[19]. These root responses have been associated with high auxin activity in the root cap. However, whether the perception and/or responsiveness to halo-stimulation is linked to root cap-derived signals has not yet been established.

Analogous to hydrotropism[20], halotropic root bending has to overcome gravity[21]. Gravisensing mainly takes place in the columellar cells of the root cap, where starch-containing plastids (amyloplasts or statoliths) settle to the bottom, triggering a cascade signaling pathway that eventually leads to downward root bending[17]. Amyloplast distribution in the root cap of excised pea plants has been implicated in the salinity-dependent regulation of gravitropic responses[22]. More crucially, salt stress was found to rapidly trigger root cap amyloplast degradation, leading to an impaired gravitropic growth response in Arabidopsis[23]. Given the central importance of the root cap not only in perceiving and transmitting environmental signals but also in involving dynamic auxin reflux loops, whether root cap-derived signals contribute to the regulation of the salt avoidance response awaits being revealed.

## Results

### The root cap-localized SMB is essential for halotropic root bending

To identify the potential genetic components involved in determining root halotropism, a T-DNA insertion population comprising 6866 confirmed T-DNA insertion lines (stock number: CS27941 from the Arabidopsis Biological Resource Center) was screened by using a modified split-agar system (see Methods; Fig. 1a; Supplementary Fig. 1). In such a system, the wild-type Col-0 seedings displayed gradually increased directional root growth away from salt-containing areas, in accompanying by the elevated NaCl gradients (Fig. 1b, c and Supplementary Fig. 2). In particular, the roots were found to initiate growth direction as early as 6 h after exposure to a 250 mM NaCl gradient, and this change became more evident at 12 h and later (Fig. 1d and Supplementary Movie 1). After the preliminary screening with the salt concentration mentioned above, a T-DNA insertion line (SALK_143526), which was previously identified to carry a T-DNA insertion at the locus of the NAC transcription factor SMB and hereafter named *smb-3*[24], was found to be almost incapable of bending away from such a salt gradient (Fig. 1b–d; Supplementary Figs. 2, 3a; Supplementary Movie 1). The *smb-3*-blocked root halotropism could be thoroughly rescued by complementation with SMB-GFP driven by its native promoter (Fig. 1b, c). These results demonstrated that the effect of T-DNA insertion in *SMB* on the loss of root sensitivity to salt is due to the lack of SMB function.

Because *smb-3* seedlings exhibited reduced root elongation, we tested whether the loss of halotropic bending in *smb-3* seedlings could be attributed to defects in root growth. Root elongation was evaluated 3 days after halo-induction. After exposure to a 250 mM NaCl gradient, the *smb-3* roots continued elongating and even entered the salt-containing area, while the Col-0 roots were observed to grow away from the salt-containing area (Supplementary Fig. 2e; Supplementary Fig. 3b). These results indicated that the failure of halotropic bending in *smb-3* does not result from root elongation defect.

To further explore whether SMB regulates root halotropism, we assessed the impact of salt exposure on the root phenotype of an inducible *SMB* transgenic line expressing rat glucocorticoid receptor (GR)-tagged SMB protein under the control of the cauliflower mosaic virus *35S* promoter (*35S:SMB-GR*). In this controllable system, the SMB-GR chimeric protein is produced constantly in the absence of added dexamethasone (DEX) and is retained in the cytoplasm, while in the presence of DEX, the fusion protein could relocate to the nucleus[25]. Exogenous application of 0.1 μM DEX to this inducible line could drive directional root growth away from a relatively low NaCl gradient (100 mM), whereas Col-0 roots still exhibited normal gravitropic growth under the same conditions (Fig. 1e, f). Further, the *35S:SMB-GR* seedlings treated with DEX and exposed to higher NaCl gradients exhibited greater root sensitivity to salt than did both the DEX-treated Col-0 and mock-treated *35S:SMB-GR* seedlings (Fig. 1e, f; Supplementary Figs. 4 and 5). These results suggested that SMB plays a positive regulatory role in root halotropism.

Furthermore, we investigated whether SMB-dependent halotropism is associated with alterations in $Na^+$ accumulation. Intriguingly, there were no apparent differences in $Na^+$ accumulation, the $Na^+/K^+$ ratio, or $Na^+$ uptake in the roots between Col-0 and *smb-3* during NaCl treatment (Supplementary Fig. 6a–c). Additionally, the biomass of whole seedlings was reduced to a similar level between Col-0 and *smb-3* seedlings subjected to salt stress (Supplementary Fig. 6d, e). The steady-state mRNA levels of *SOS1*, *SOS2* and *SOS3*, three key elements involved in salt perception and uptake[26], were comparable in the root tips of Col-0 and *smb*−3 after 2 h of NaCl treatment (Supplementary Fig. 6f). Taken together, these findings provide no evidence that SMB-mediated root halotropism is due to a change in the cellular $Na^+$ concentration.

Additionally, the transcript abundance of the *SMB* gene did not significantly increase in the Col-0 seedlings when they were exposed to a 250 mM NaCl gradient (Supplementary Fig. 7a). Furthermore, in vivo analysis of an SMB transcriptional reporter line (*pSMB:nls-GFP*) and a translation reporter line (*pSMB:SMB-GFP/smb-3*) both revealed that there was no increase in but comparable fluorescence intensity between the two sides of LRC over 4 h of exposure to a 250 mM NaCl gradient (Supplementary Fig. 7b–e). It appears that the *SMB* expression patterns are less influenced by the salt gradient during the early phase of halo-stimulation.

### SMB-mediated halotropism is uncoupled to an attenuated columellar amyloplast-dependent gravitropic growth response

It has been increasingly accepted that root halotropism is accompanied by the transient repression of root gravitropism[27]. Compared with the WT seedlings, the *smb-3* seedlings had a reduced root-wave and greater gravitropic index (Supplementary Fig. 8a, b). This finding raises the possibility that SMB-regulated root halotropism is due to a decreased gravitropic response. To investigate this phenomenon, we tested the root gravitropic response of *smb-3* seedlings to salt treatments. In the absence of salt treatment, the roots of these seedlings exhibited gravitropic responses similar to those of Col-0 seedlings upon gravti-stimulation (Supplementary Fig. 8c–e). Attenuation of root gravitropism in Col-0 was observed at as low as 75 mM NaCl and was greatest at 150 mM NaCl, whereas smb-3 still showed normal gravitropism at 75 and 100 mM NaCl and slightly decreased gravitropism at higher salt concentrations (≥125 mM) (Supplementary Fig. 8c–e). These results indicate that SMB is not required for gravitropism but plays a specific role in root halotropism. Based on these findings, we propose that SMB may act downstream of salt stress signaling and that salt-activated root halotropism occurs prior to gravitropism.

Root tropism is related to amyloplast sedimentation in root cap columella cells, which is reduced under high salinity, leading to the compromised ability of roots to sense gravity[23]. We thus probed whether amyloplast sedimentation is involved in SMB-dependent

halotropism. The amyloplasts in the columella were visualized by Lugol's staining and modified propidium iodide staining (mPS-PI). In line with the findings of a previous report[23], the amount of amyloplasts in columellar cells at non-gradient/ homogeneous salt-containing medium substantially decreased with increasing concentrations of NaCl (Supplementary Fig. 9). In contrast, the columella of *smb-3* sustained higher content of amyloplasts than did those of Col-0 under both mock and salt treatments (Supplementary Fig. 9). To our surprise,

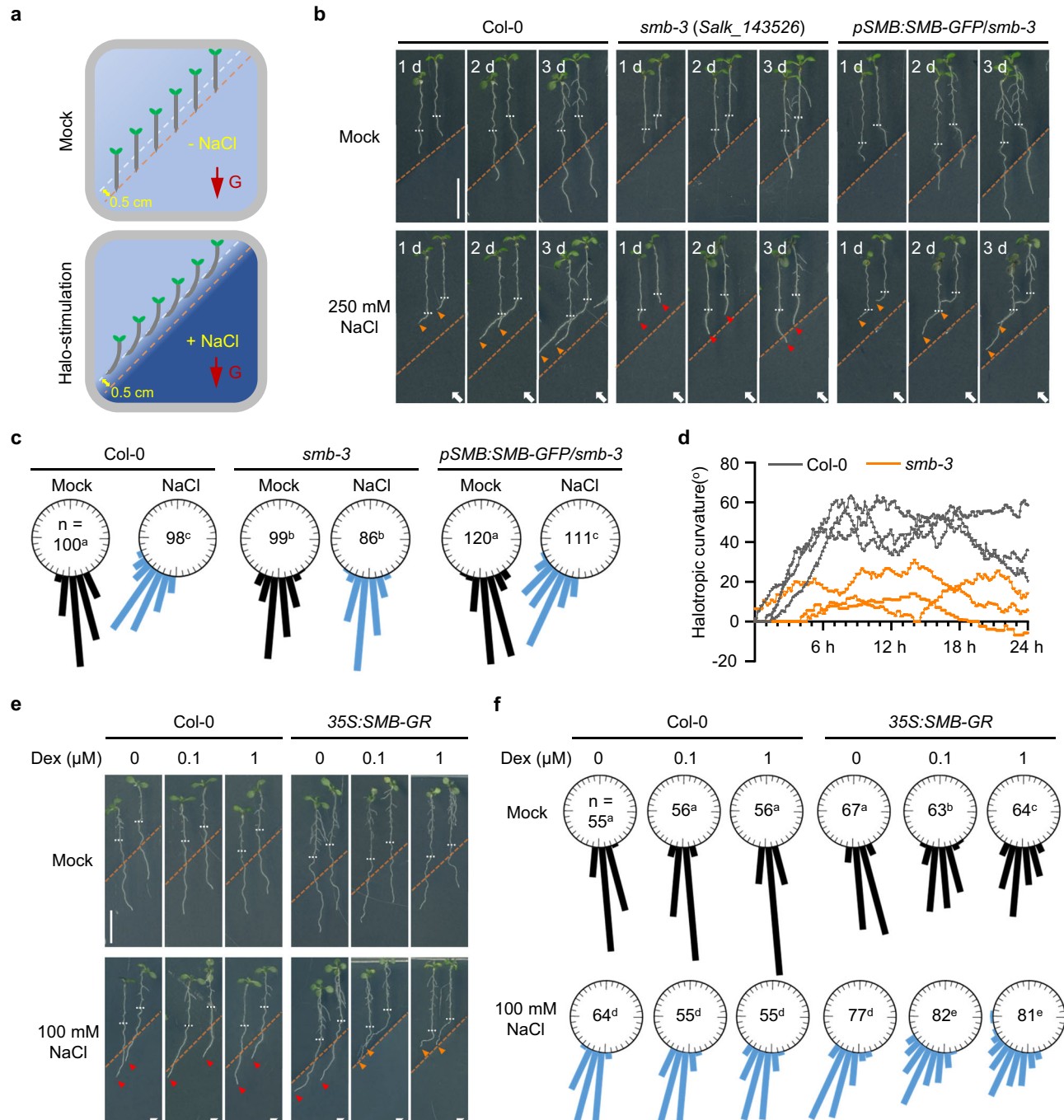

**Fig. 1 | SMB is required for root halotropic bending. a** Schematic diagram for the root halotropic bending assay. Three-day-old seedlings were transferred to 1/2 MS split-agar medium containing mock (NaCl-free) or indicated NaCl concentrations at the bottom right side. The root tips were placed 0.5 cm (white dashed line) above the mock-salt or mock-mock boundary (orange dashed line). G indicates the direction of gravity. **b** Halotropic root responses of Col-0 and *smb-3* seedlings that were transferred to split-agar medium with or without 250 mM NaCl for 3 days. Scale bar, 1 cm. **c** Quantification of the percentage of roots in angle categories of 10 degrees after 1 day of treatment, with the number of roots measured. 0° equals vertical. The black and blue bars represent the Mock and NaCl treatments, respectively. **d** The curvature of halo-stimulated Col-0 (gray) and *smb-3* (orange) roots was measured at 5-min intervals over 24 h of halo-stimulation (three seedlings of each genotype), as also visualized in Supplementary Movie 1. **e, f** Halotropic root responses of Col-0 and *35S:SMB-GR* seedlings that were transferred to split-agar medium with or without 100 mM NaCl in the absence or presence of DEX for 3 days (**e**). Scale bar, 1 cm. Halotropic root curvature was quantified and shown in **f**. In **b** and **e**, white arrows indicate the direction of NaCl diffusion; white dotted lines represent the initial location of the root tip when the salt gradient was created; orange dotted line represents the mock-mock or mock-salt boundary; orange arrows represent the occurrence of halotropic root bending; and red arrows indicate the absence of halotropic root bending. In **c** and **f**, *n* indicates the number of independent seedlings, and alphabets denote significant differences (*P* < 0.05, two-way ANOVA by Tukey's test).

both sets of statolith staining results showed that compared with those in the corresponding control treatments, no decrease in the amount of amyloplasts was detected in either the Col-0 or *smb-3* columella 24 h after exposure to 150 or 250 mM salt gradients (Fig. 2a–c; Supplementary Fig. 10). These contrasting results could be explained by the fact that the actual salt concentration experienced by the root cap cells was not enough to trigger amyloplast degradation when exposed to diagonal NaCl gradients. Galvan-Ampudia et al. [10] previously reported that root tips experience a gradual increase of NaCl, raising up to about 26% of the initial concentration of the salt-containing medium at 24 h of halo-stimulation. Hence, when the seedlings were placed in the maximal salt gradient (at 250 mM), an estimated 65 mM NaCl concentration was confronted by the root cap cells at 24 h of halo-stimulation. This finding is in agreement with the observations that there was no apparent degradation of amyloplasts in Col-0 or *smb-3* on solid media containing low concentrations of NaCl (below 75 mM) during 24 h of treatment (Supplementary Fig. 9). Altogether, these results imply that halotropic root responses mediated by SMB are less closely linked to alterations in amyloplast sedimentation.

To further validate this possibility, analysis of genetic interactions between *SMB* and two starch synthesis genes *STARCH SYNTHASE 4* (*SS4*) and *ADP GLUCOSE PYROPHOSPHORYLASE 1* (*ADG1*)[28] in the regulation of root halotropism was performed. In agreement with previous observations[29], the amount of amyloplasts in the columella was largely reduced in *ss4-3*, and fully annulled in *adg1-1*, regardless of the absence or presence of salt treatment (Fig. 2a–c). Further, both mutations resulted in not only lower sensitivity to gravity but also greater sensitivity to the salt gradient (Fig. 2d, e; Supplementary Figs. 11–13). These two opposite findings may hint that gravitropism antagonizes halotropic root bending. The enhanced halotropic root bending in *ss4-3* and *adg1-1* is therefore likely to be caused by impaired gravitropism, but not enhanced halotropism. Intriguingly, neither the *ss4-3* nor the *adg1-1* mutation could alter the root response of *smb-3* to a high level of salt gradient (250 mM NaCl; Fig. 2d, e), though the amount of amyloplasts in the *ss4-3 smb-3* and *adg1-1 smb-3* double mutants was dramatically reduced to levels comparable to those in the *ss4-3* and *adg1-1* single mutants (Fig. 2a–c). These results firmly support the conclusion that SMB drives halotropic root bending unlikely via down-regulating root-cap-localized statolith and related gravitropic growth responses. Again, amyloplast-dependent gravity signaling pathways appear not to be involved in SMB-mediated root halotropism.

### SMB regulates asymmetric auxin distribution in the root cap to promote halotropism

While root cell death was proposed to contribute to adaptive growth in coping with a saline environment[30], SMB is capable of stimulating the differentiation of young root cap cells and transcriptionally activating programmed cell death (PCD) in LRC cells at the distal end of the root meristem[31]. Indeed, the PCD of LRC cells was discontinued in *smb-3* seedlings, and more LRC cells were observed in the root tips, irrespective of the absence or presence of a salt gradient (Supplementary Fig. 14). However, compared with those of mock-treated Col-0 seedlings, no increased cell death around the root tips of Col-0 seedlings was observed following treatment with a 250 mM NaCl gradient (Supplementary Fig. 14). This result implies that halotropism appears not to require SMB-controlled developmental PCD.

Recent evidence implicated that decreased gravitropism by down-regulating auxin biosynthesis and signaling pathways can enhance halotropic root bending[32]. To test whether SMB mediates halotropism via the auxin-related pathway, we challenged halostimulated roots with the specific auxin antagonists yucasin, α-(phenylethyl-2-one)-indole-3-acetic acid (PEO-IAA), and ₗ-kynurenine (ₗ-Kyn). The application of yucasin, PEO-IAA or ₗ-Kyn caused an agravitropic root response in both Col-0 and *smb-3* seedlings (Supplementary

Fig. 15a, b). Remarkably, all the tested auxin antagonists accelerated the root halotropic bending of Col-0, and restored the root halotropic response of *smb-3* when exposure to a NaCl gradient (Supplementary Fig. 15c–e). In line with the aforementioned notion that the loss of halotropic bending in *smb-3* seedlings did not involve root elongation defects, the halotropic responses of auxin antagonist-treated *smb-3* seedlings were restored independently of primary root inhibition (Supplementary Fig. 15e). Halotropic enhancement and/or reversion were likely correlated with a significant decrease in the gravitropic response. Consistently, a dramatic decrease in auxin-responsive *DR5rev:3xVENUS-N7* expression was observed in the root tip of the starch synthesis mutant *ss4-3*, which displayed accelerated root halotropic bending and decreased gravitropic growth (Supplementary Fig. 15f, g). These results further demonstrated the involvement of auxin signaling in the SMB-dependent halotropic root response.

Lateral auxin gradient occurred at the outer layer tissue (e.g., LRC and epidermis) of the root tip is required to stir up the halotropic root bending[10]. Hence, we examined if the stimulative effect of SMB on halotropic root bending is linked to auxin redistribution, using the sensitive auxin response reporter *DR5rev:3xVENUS-N7*. During the exposure of Col-0 roots to a 250 mM NaCl gradient, *DR5* expression progressively increased in LRC cells at the side of the root opposite to the NaCl source, as early as 2 h preceding the root halotropic bending, while it decreased at the side of the root nearest to the salt (Fig. 3a–c; Supplementary Fig. 16; Supplementary Movie 2). This indicated a higher auxin signaling in these tissues opposite to the high salt concentration, which was consistent with previously reported results[10]. Notably, the increase in the DR5 fluorescence signal in the non-salt-exposed side and decrease in the salt-exposed side lasted until 18 h after halo-stimulation (Fig. 3c–d and Supplementary Movie 2). Afterwards, this transient change became less pronounced, and gravity signaling appeared to predominate over halotropic signaling as the Col-0 roots showed the trend to grow downwards (Fig. 3c–d and Supplementary Movie 2). In contrast, *smb-3* exhibited dramatically decreased *DR5* expression at the both sides of LRC and epidermis, and also dismissed the asymmetric distribution of the DR5 signal in these tissues during 24 h after halo-stimulation (Fig. 3c and Supplementary Movie 3). As a result, the halotropic bending of the roots of Col-0 seedlings during the 24 h of halo-stimulation almost completely disappeared in the *smb-3* seedlings (Fig. 3d and Supplementary Movies 2 and 3). The above results suggested a regulatory role of SMB in the local accumulation and asymmetric distribution of auxin in outer layer root tissues, which is required for root halotropism.

Because SMB is strictly distributed at the root cap (Supplementary Fig. 7b–e)[33]. This promotes us to test whether the effect of SMB on halotropism is relevant with root cap-derived auxin dynamics. For this purpose, a line expressing *indole-3-acetamide hydrolase* (*iaaH*)[34] under the control of the SMB promoter (*pSMB:iaaH*) was generated, in which indole-3-acetamide (IAM) is converted to IAA in the root cap. No visible change in LRC death was observed in the root tips of the *pSMB:iaaH* line in the absence or presence of IAM when exposed to a 250 mM NaCl gradient (Supplementary Fig. 17a, b), whilst *pSMB:iaaH* had an elevated auxin signal in the root tip and no root halotropic bending in response to the NaCl gradient upon IAM addition (Fig. 3e, f; Supplementary Fig. 17c–i). This finding is in line with the blocking effect of the addition of naphthalene-1-acetic acid (NAA) (Fig. 3g, h and Supplementary Fig. 18) or 3-indoleacetic acid (IAA)[10] on asymmetrical auxin distribution and halotropic response. These results demonstrated that SMB mediates halotropism via the local regulation of auxin redistribution in the root cap.

### Coupling spatiotemporal patterns of AUX1 to auxin redistribution can explain SMB action in the control of root halotropism

Auxin uptake and redistribution in LRC cells requires the auxin influx carrier AUX1, whose mutation causes an agravitropic root response[16].

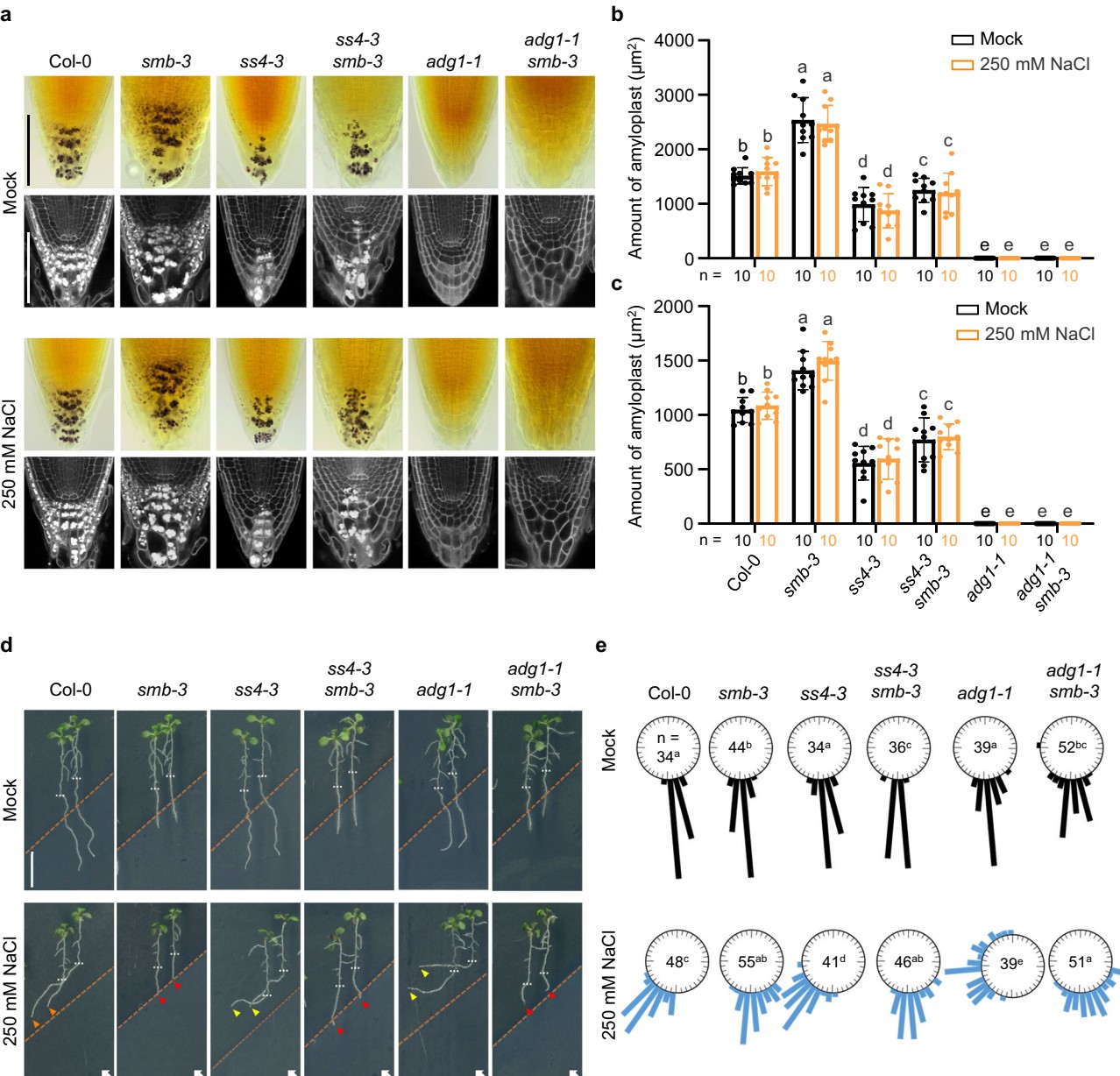

**Fig. 2 | Genetic analysis of root cap starch synthesis genes in the regulation of SMB-dependent halotropic bending. a–c** Amyloplast staining and quantification of the root apexes of Col-0, *smb-3*, *ss4-3*, *adg1-1*, *ss4-3 smb-3* and *adg1-1 smb-3* seedlings that were transferred to split-agar medium with or without 250 mM NaCl for 6 h (**a**). Scale bar,100 μm. Amyloplasts of the root apex were quantified via Lugol's staining (**b**) and mPS-PI staining (**c**). Values are means ± SD (*n* = 10 seedlings). **d**, **e** Halotropic root responses of Col-0, *smb-3*, *ss4-3*, *adg1-1*, *smb-3*, *ss4-3 smb-3* and *adg1-1 smb-3* seedlings that were transferred to split-agar medium with or without 250 mM NaCl for 3 days (**d**). White arrows indicate the direction of NaCl diffusion; white dotted lines represent the initial location of the root tip when the salt gradient was created; orange dotted line represents the mock-mock or mock-salt boundary; orange arrows indicate the occurrence of halotropic root bending; red arrows indicate the absence of halotropic root bending; yellow arrows indicate accelerated root halotropic bending in the indicated genotypes, relative to Col-0 (orange arrows); and red arrows indicate compromised root halotropic response. Scale bar, 1 cm. Halotropic root curvature was quantified and shown in **e**. In **b**, **c** and **e**, *n* indicates the number of independent seedlings, and alphabets denote significant differences (*P* < 0.05, two-way ANOVA by Tukey's test).

We found that the *aux1-21* knock-out mutant exhibited substantially reduced *DR5* expression at both sides of the LRC but enhanced root halotropism compared to Col-0 when exposed to NaCl (Fig. 4a–d and Supplementary Fig. 19), indicating that the loading of auxin into the LRC is required for halotropism. Interestingly, the *aux1-21 smb-3* double and *aux1-21* single mutants showed comparable gravitropism defects in the absence of NaCl and identical enhanced root halotropic responses upon halo-stimulation (Fig. 4a–d; Supplementary Figs. 19–21). These results indicate that AUX1 may be involve in SMB-mediated auxin redistribution and root halotropism.

To probe the molecular link between SMB and AUX1, we first searched for potential SMB binding sites through a released database of the *Arabidopsis* cistrome using DNA affinity purification sequencing (DAP-seq) technology[35]. The DAP-seq data showed that the 2710- to 2197-bp promoter region of *AUX1* harbors a binding site (named *cis-element 1*) for SMB (Fig. 4e and Supplementary Fig. 22a). Apart from the binding site determined by DAP-seq, bioinformatics analysis revealed that this promoter region (−1970 to −843 bp) contains two conserved motifs (*cis*-elements 2 and 3; Fig. 4e, f). Subsequently, we checked for an interaction between SMB and

these *cis*-elements in the promoter region of *AUX1* via yeast one-hybrid (Y1H) and chromatin immunoprecipitation-polymerase chain reaction (ChIP-PCR) experiments. Both the Y1H and ChIP-PCR assays confirmed that SMB can bind only to *cis*-element 1

in vitro and in vivo (Fig. 4g, h and Supplementary Fig. 22b). Critically, *AUX1* expression was partially reduced in the root tips of *smb-3* seedlings in the presence or absence of the salt gradient compared with that in the roots of Col-0 seedlings (Fig. 4i). Collectively, these

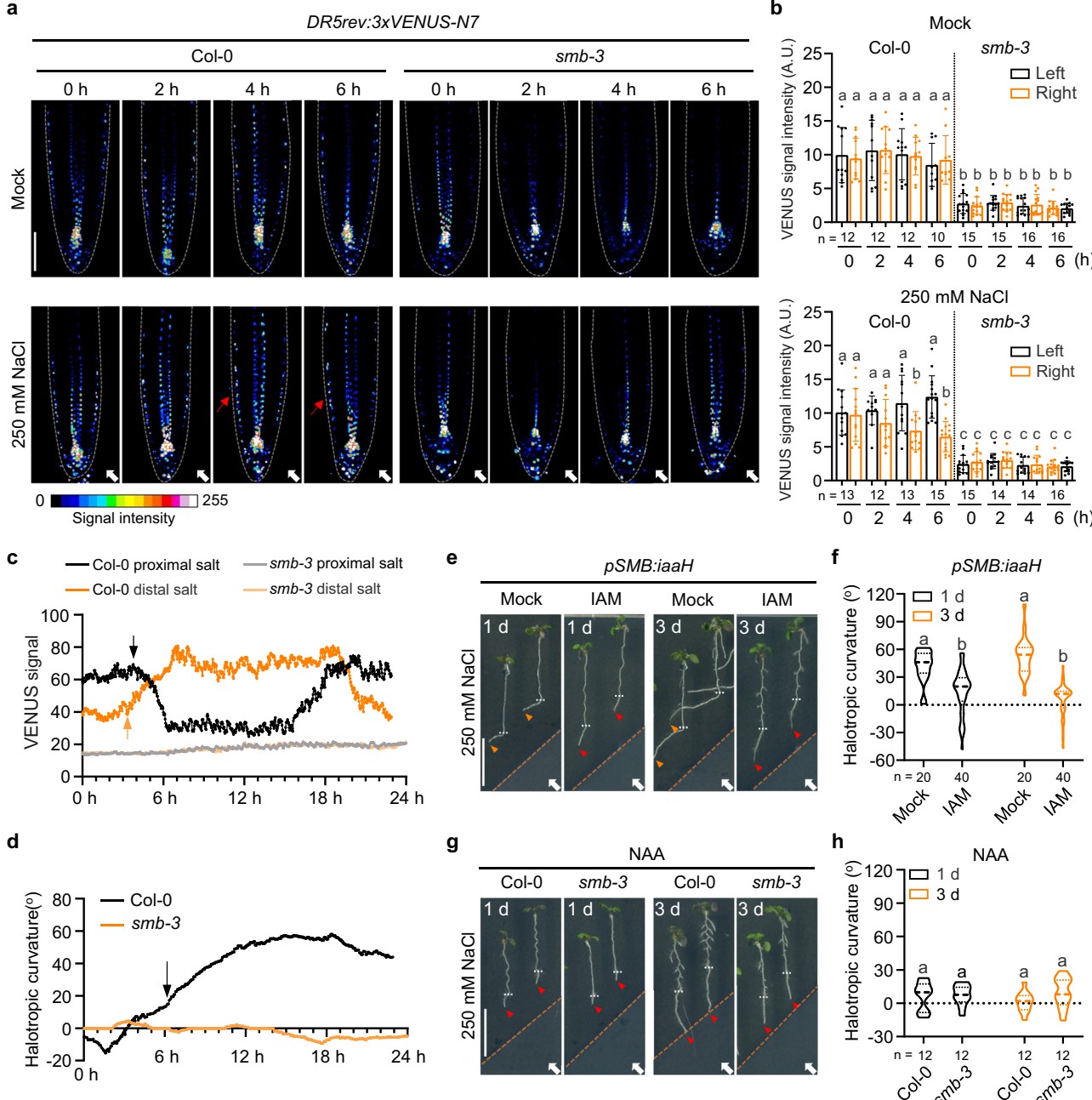

**Fig. 3 | SMB regulates asymmetric auxin distribution in the root tip under halostimulation. a, b** Confocal images of DR5rev:3xVENUS-N7 signal in LRC and epidermal cells of Col-0 and *smb-3* seedlings that were transferred to split-agar medium with or without 250 mM NaCl for 6 h (**a**). White arrows indicate the direction of NaCl diffusion, while red arrows highlight a greater DR5rev:3xVENUS-N7 signal intensity at root tissues. Dashed white lines denote the tentative outline of the root tip. Scale bar, 100 μm. The fluorescence intensity of DR5rev:3xVENUS-N7 in the LRC and epidermis distal (left) or proximal (right) to mock/salt gradient was quantified (**b**), values are means ± SD. Time-lapse analysis of DR5rev:3xVENUS-N7 signal intensity (**c**) and root curvature (**d**) of Col-0 and *smb-3* seedlings that were transferred to split-agar medium with or without 250 mM NaCl. DR5 signal intensity and root curvature were measured at 2-min intervals over 24 h, as visualized in Supplementary Movies 2 and 3. The black and orange arrows in **c** indicate the time points at which the auxin concentration began to decline at the proximal salt region

of Col-0 root tip and increase at the distal salt region, respectively. The black arrow in **d** shows the time point at which the Col-0 root tip began to bend away from salt gradient. Halotropic root response of *pSMB:iaaH* seedlings after 1 day and 3 days of halo-stimulation in the absence or presence of 1 μM IAM (**e**), and root curvature was quantified (**f**). Scale bar, 1 cm. Halotropic root response of Col-0 and *smb-3* after 1 day and 3 days of halo-stimulation in the presence of 0.3 μM NAA (**g**), and root curvature was quantified (**h**). Scale bar, 1 cm. In **e** and **g**, white arrows indicate the direction of NaCl diffusion; white dotted lines represent the initial location of the root tip when the salt gradient was created; orange dotted lines represent mock-mock or mock-salt boundary; orange and red arrows represent the presence and absence of halotropic root bending, respectively. In **b**, **f** and **h**, n indicates the number of independent seedlings, and alphabets indicate significant differences (*P* < 0.05, two-way ANOVA by Tukey's test).

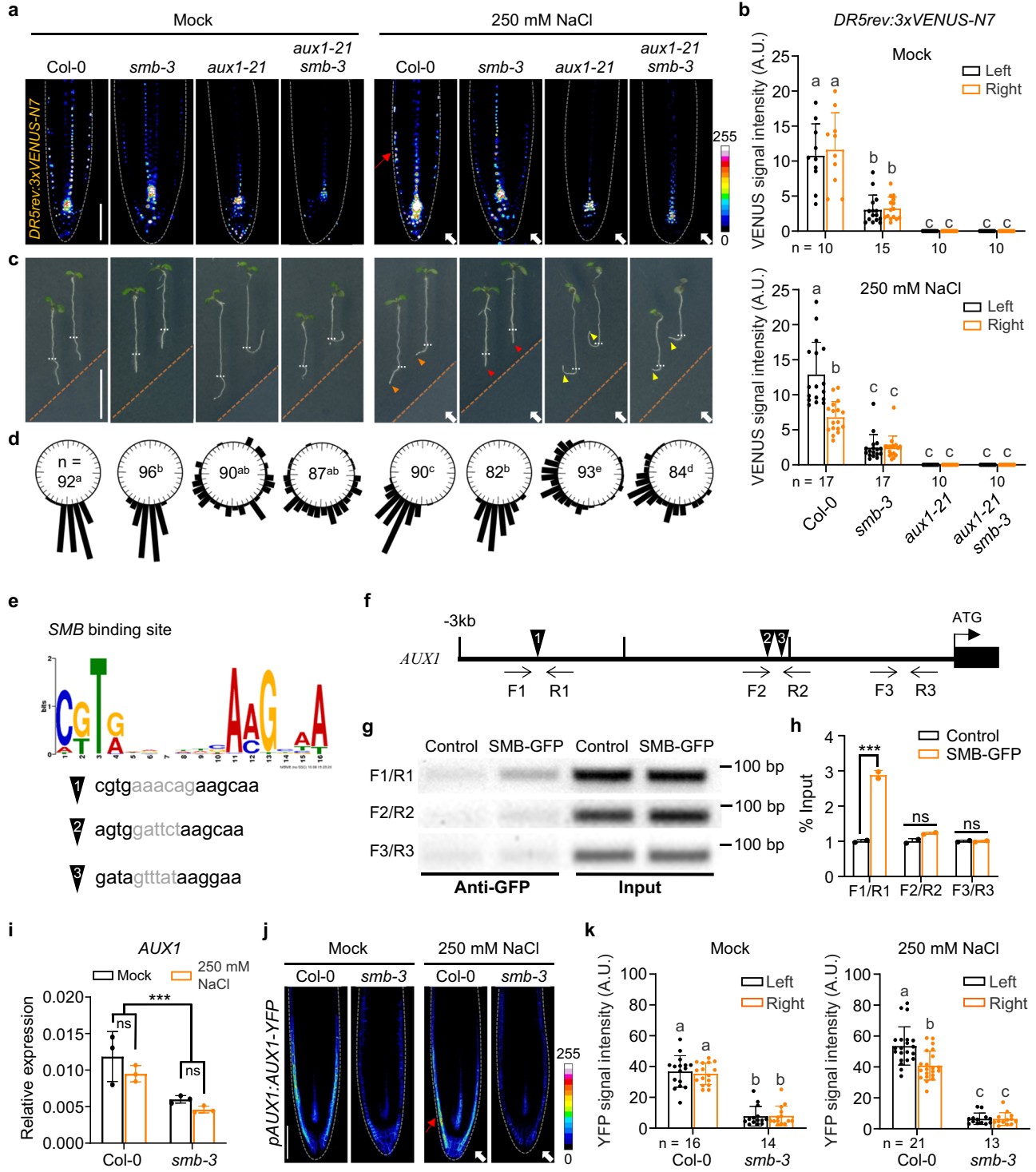

results demonstrated that SMB could bind to the promoter of *AUX1* and maintain its basal expression.

Further, the Col-0 seedlings carrying the *pAUX1:AUX1-GFP* construct exhibited higher AUX1-GFP abundance at the non-salt-exposed side of the roots than at the salt-exposed side of the roots after halo-stimulation, consistent with previous observations[12]. However, compared with Col-0 seedlings, the *smb-3* seedlings carrying *pAUX1:AUX1-GFP* showed dramatically lower level of AUX1-GFP at the both sides of the LRC, regardless of the absence or presence of the NaCl gradient (Fig. 4j, k). This local repression of AUX1 in *smb-3* most likely causes LRC auxin deficiency, and thus is insufficient to trigger asymmetric auxin transport following the halo-stimulation. These evidences

collectively support the hypotheses that SMB operates via halotropism per se, and that SMB drives the expression of *AUX1* specifically in the LRC to stimulate halotropism instead of altering the AUX1-dependent gravitropic response.

## Discussion

In conclusion, our findings uncover a decisive role of SMB in root halotropism, which is related with the transcriptional regulation of local *AUX1* expression in LRC. It is noteworthy that although *SMB* expression is not asymmetrically induced by the salt gradient, it is essential for basal expression of the *AUX1* gene at the both sides of the LRC, and contributes to the establishment of asymmetrical *AUX1*

**Fig. 4 | SMB regulates the transcription of *AUX1* to monitor auxin redistribution in root tip and root halotropism. a, b** Confocal images of DR5rev:3xVENUS-N7 signal in the root tips of indicated genotypes during 6 h of halo-stimulation. Scale bar, 100 µm. The fluorescence intensity of DR5rev:3xVENUS-N7 in the LRC and epidermis distal (left) or proximal (right) to mock/salt gradient was quantified (**b**). Halotropic root response of indicated genotypes after 24 h of halo-stimulation (**c**), and root curvature was quantified (**d**). White arrows indicate the direction of NaCl diffusion; white dotted lines represent the initial location of the root tip when the salt gradient was created; orange dotted line represents mock-mock or mock-salt boundary; yellow arrows indicate accelerated root halotropic bending at indicated genotypes, relative to that of Col-0 (orange arrows); red arrows indicate a compromised root halotropic response. Scale bar, 1 cm. **e** Predication of SMB binding sites based on the plant cistrome database[35]. **f** Schematic representation of putative SMB-binding sites (#1-3) in *AUX1* promoter with three primer sets designed for ChIP–PCR. **g, h** In vivo binding test of SMB to putative *cis*-elements by ChIP–PCR.

The nuclei extracted from *smb-3* plants complemented with the *pSMB:SMB-GFP* construct (SMB-GFP) and *p3SS:GFP* (control) were immunoprecipitated with anti-GFP antibody. PCR was performed using the primers indicated in **g**, and the SMB enrichment relative to the input was quantified ($n = 2$ independent biological replicates) (**h**). **i** qRT–PCR analysis of *AUX1* transcripts from Col-0 and *smb-3* root tips after 6 h of halo-stimulation ($n = 3$ independent biological replicates). Confocal images (**j**) and quantification (**k**) of AUX1-YFP signal intensity in Col-0 and *smb-3* root tips after 6 h of halo-stimulation. Scale bar, 100 µm. AUX1-YFP signal intensity at each side of LRC was quantified. In **a** and **j**, the dashed white lines denote the tentative outline of the root tip; white arrows indicate the direction of NaCl diffusion; red arrows highlight a higher *DR5rev:3xVENUS-N7/AUX1-YFP* signal intensity at root tissues. In **b** and **k**, data values are means ± SD; n indicates number of independent seedlings, and alphabets indicate significant differences ($P < 0.05$, two-way ANOVA by Tukey's test). In **h** and **i**, values are means ± SD; statistical analysis was performed with two-tailed Student's *t* test (\*\*\**P* < 0.001; ns not significant).

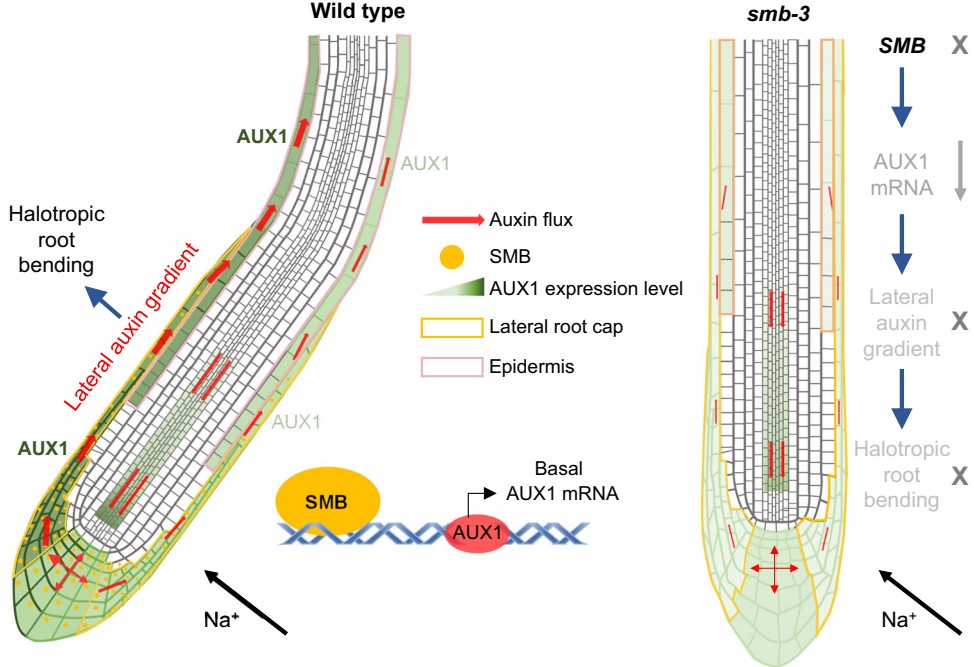

**Fig. 5 | Schematic model for SMB-dependent root halotropism.** In this model, the root cap-localized NAC transcription factor SMB can bind to the promoter of the auxin influx carrier-encoding gene *AUX1*, and positively regulate the expression of *AUX1* in the root cap, in turn activating the halotropic root response. *AUX1* is highly expressed in the lateral root cap (LRC) and epidermis and is required for the establishment of a lateral auxin gradient in response to gravity. Upon halo-stimulation, SMB can activate *AUX1* expression to facilitate auxin accumulation in the LRC and epidermis, allowing the establishment of a lateral auxin gradient to provoke a halotropic root response. In *smb-3*, the knock-out mutant of *SMB*, *AUX1* expression in the LRC and epidermis was reduced, accompanied by low auxin accumulation and disruption of the lateral auxin gradient, eventually leading to loss of the halotropic root response. However, SMB does not directly act on the establishment of lateral auxin gradient, which might be regulated by uncharacterized factors.

expression and a lateral auxin gradient in the LRC, which is required for the activation of the root halotropic response. Uncharacterized pathways independent of SMB further activate unilateral *AUX1* expression in the LRC at the side of the root opposite the high salt gradient. These combined effects may provide adequate transport proteins for auxin movement, suggesting that multiple regulatory factors in the root cap are required for synergistically governing root halotropic bending in an auxin-dependent asymmetrical regulation (Fig. 5). This is evident through the observations of *smb-3* seedlings with low *AUX1* transcript abundance and auxin levels, which fail to establish asymmetrical auxin distribution and eventually lead to abortion during the execution of root halotropism. Consistently, it has been previously revealed that AUX1-dependent auxin dynamics in the root tip are the key to driving the halotropic response[11,12].

Our results also suggested that starch synthesis in the root cap columella is crucial for root halotropism. Analysis of starch synthesis mutants and *aux1* mutants with substantial loss of the DR5 signal in LRC and enhanced halotropism also highlights the importance of auxin in the regulation of the root response to salt exposure downstream of amyloplast sedimentation. Auxin has been previously suggested to regulate starch synthesis in the root cap[29], while our results further showed that starch synthesis is also essential for auxin accumulation in the LRC, indicating mutual regulation between auxin and starch synthesis. However, starch synthesis is not involved in the regulation of the root halotropic response by SMB, as indicated by the following observations. First, blocking starch synthesis in *smb-3* reduced the amyloplast contents in the root cap, but did not alter the response of the *smb-3* root to the salt gradient. Second, the

amyloplast content was also severely reduced in *smb-3* seedlings when exposed to salt concentrations that exceed 100 mM. Thus, starch synthesis and SMB might act through distinct molecules or pathways to modulate auxin distribution in LRC.

Overall, the contribution of SMB to basal *AUX1* expression and auxin redistribution in the LRC is essential for driving halotropic root bending, thus presenting that the spatiotemporal regulation of the SMB-AUX1-auxin signaling module in the root cap is a central hub in determining root halotropism. Our findings uncover a previously unidentified role for root cap function in halotropism-specific pathways, and thus identifying and engineering SMB-associated regulatory genes in the root cap could pave the way for improving plant root adaptation to salt stress. These findings also highlight the central roles of the root cap at the crossroads of multiple tropisms including the newly described halotropism in complex and fluctuating environments.

## Methods

### Plant materials, growth conditions, and root elongation measurements

The *Arabidopsis thaliana* plants used in all the experiments were the Columbia-0 (Col-0) ecotype. The previously described transgenic lines used here were *35S:SMB-GR*[24], *pAUX1:AUX1-YFP*[36], *DR5rev:3xVENUS-N7*[37], *DR5:GUS*[38], *DII:VENUS*[39] and *pSMB:SMB-GFP/smb-3*[24]. The mutant seeds of *ss4-3* (SALK_096130)[40] and *adg1-1* (CS3094)[41] were obtained from Guanghui Xiao (Shaanxi Normal University). The *smb-3* (SALK_143526; 21), *aux1-21* (CS9584)[42] mutants and a confirmed T-DNA insertion library at the coverage of 6866 genes (CS27941) were obtained from the Arabidopsis Biological Resource Center. Homozygotes were identified using sequence information obtained from the SIGnAL website at http://signal.salk.edu.

The double knockout mutant plants *ss4-3 smb-3*, *adg1-1 smb-3* and *aux1-21 smb-3* were generated by crossing *smb-3* plants (pollen acceptors) with the respective mutant lines (pollen donors). A double mutant was identified in F2-segregating populations by PCR and/or Sanger sequencing-based genotyping with gene-specific primers and T-DNA specific primers as shown in Supplementary Table 1. For the *ss4-3 and adg1-1* mutations, the progenies of the crosses were screened for either reduced-starch or starch-free phenotypes by iodine staining. To generate *smb-3* plants harboring the *DR5rev:3xVENUS-N7* or the *pAUX1:AUX1-YFP* transgenes, *smb-3* plants were crossed with wild-type plants harboring *DR5rev:3xVENUS-N7* or *pAUX1:AUX1-YFP*.

Seeds were surface-sterilized then sown on solid half-strength MS medium (1% sucrose, 0.05% 2-(N-morpholino) ethanesulfonic acid (MES), pH 5.7, 0.8% agar). Plates with seeds were stratified at 4 °C in the dark for two days. The seeds were cultured vertically at 22 °C under continuous white light (PAR of 100 to 120 μE m$^{-2}$ s$^{-1}$). After 5 days of germination, 3-day-old seedlings from the indicated genotypes were used for experiments unless otherwise specified. The root elongation was calculated as the difference between the length of the primary roots after and before transfer for the indicated times. The primary root length was quantified using ImageJ software.

### Plasmid construction and plant transformation

To generate the *pSMB:nls-3xVENUS* and *pSMB:iaaH* constructs, the *SMB* promoter fragments upstream of the coding sequence (~4 kb) were amplified from genomic DNA, subcloned and inserted into pDONRP4P1R by Gateway cloning, and finally fused with nuclei-tagged 3xVENUS (nls-3xVENUS) or iaaH in a destination vector (pB7m24GW). *pSMB:nls-3xVENUS* and *pSMB:iaaH* transgenic plants was obtained by Agrobacterium-mediated transformation of the respective expression clones in the Col-0 background. Furthermore, plants containing the *pSMB:iaaH* transgene were crossed with *DR5:GUS* or *DII:VENUS* to generate *pSMB:iaaH/DR5:GUS* and *pSMB:iaaH/DII:VENUS* double transgenic plants respectively. Homozygous plant lines were finally used for the experiment and analysis.

### Halotropic root response assay

The analysis of halotropic root bending was conducted via a vertical split-agar assay[10] with slight modifications. Briefly, diagonal NaCl gradients were generated by cutting the lower part of the plate with a sterile thin glass plate and replacing the medium with fresh melted 1/2 MS medium supplemented with different concentrations of NaCl. The plates were left horizontally for 30 min to allow gel solidification. Then, three-day-old seedlings from the indicated genotypes were transferred to the split-agar medium for halo-stimulation. For the halotropic root bending assay with auxin-related compounds, 10 μM PEO-IAA, 10 μM yucasin, 1 μM L-Kyn, 0.3 μM NAA and 1 μM IAM were added to 1/2 MS medium supplied with or without 250 mM NaCl and used for the split-agar assay. The root phenotype was recorded every 24 h after halo-stimulation with a scanner (EPSON XL11000). The capacity of the roots to grow away from the salt gradient was conceptualized as a bending angle as shown in Supplementary Fig. 1. For each root, the bending angle was quantified by first drawing a straight line connecting the locations of the root apex upon halo-stimulation (i.e., the start/zero time point) and after the indicated number of days. Second, the angle was measured between the drawn line and the gravity vector. The bending angle was assessed after one and three days of growth. All the data were measured and analyzed using ImageJ software.

### Root gravitropic bending assay

For observation of the root wave, the vertical growth index (VGI) was defined as the ratio between the root tip ordinate and the root length[43]. Three-day-old seedlings geminated on 1/2 MS were used for the experiment.

For observation of the gravity response, seedlings of the indicated genotypes were transferred to plates containing 1/2 MS medium supplied with or without NaCl, and the plates were then rotated by 90 degrees relative to the original vertical position. Root growth and direction were monitored by scanning every 24 h. In the homogeneous salt condition, the seedlings were transferred to 1/2MS medium supplemented with the indicated concentrations of NaCl.

For the gravitropic root bending assay with auxin inhibitors, the seedlings were transferred to mock, 10 μM PEO-IAA, 10 μM yucasin and 1 μM L-Kyn-supplemented 1/2 MS medium plates, which were then rotated by 90° for the indicated times. The bending angles of the root tips away from the horizontal direction were measured and analyzed using ImageJ software[23].

### Ion determination and net Na$^+$ flux assay

The roots of the *Arabidopsis* seedlings were harvested and dried at 80 °C for 24 h. Then, dry samples were digested in 1 mL of nitric acid at 90 °C for 12 h, diluted to 5 mL with distilled water. The Na$^+$ and K$^+$ contents were measured using an inductively coupled plasma–optical emission spectrometry instrument (PerkinElmer, USA)[44].

A BIO-IM Series NMT Physiolyzer ® system (YoungerUSA) was used to determine the net Na$^+$ flux in the root tip based on the methods of a previous study[45]. The roots of Col-0 and *smb-3* seedlings were placed in Petri dishes containing 20 mL of liquid medium supplemented with 100 mM NaCl. The surface of the roots was cleaned gently with a soft brush in time to prevent surface attachments from affecting the experimental results. The net Na$^+$ flux was measured along the root tip, concentrating in the following zones: 0, 150 and 300 μm from the root cap junction. The microelectrodes were positioned 0 ± 2 μm away from the samples by the computer-controlled NMT system, and each position of the sample was measured for 3–5 min.

### Confocal microscopy and quantification

A Leica SP8 laser-scanning microscope was used for fluorescence imaging of the *Arabidopsis* roots. To image propidium iodide (PI)-stained roots, the seedling roots were priorly treated with 2 μg/mL PI for 5 min, then washed with water, and finally transferred to slides for

confocal imaging. The excitation (ex) and detection (em) wavelengths for the different fluorescent dyes and proteins were as follows: GFP (ex: 488 nm, em: 505–555 nm), VENUS/YFP (ex: 514 nm, em: 525–555 nm), and PI (ex: 550 nm, em: 600–640 nm).

The quantification of fluorescence signal intensity was performed in two regions of the root tip with contrasted exposures to the applied salt stress. First, the proximal region was defined as the right side of the root tip, that is, facing the salt gradient (or the mock). Second, the opposite side (left) of the root tip was named as the distal region. At both sides of the root tip, the fluorescence was measured in the LRC and epidermal cell layer. The fluorescence signals of *pSMB:nls-3xVE-NUS*, *pSMB:SMB-GFP/smb-3*, *DR5rev:3xVENUS-N7*, and *pAUX1:AUX1-YFP* were quantified within the distal region and the proximal region. The fluorescence intensity ratios were obtained by comparing *DR5rev:3x-VENUS-N7* fluorescence intensity between the distal region and the proximal region.

Quantification of the LRC cell number was performed for individual roots. Measurements were performed on a median confocal plane image along individual LRC cell files, starting from the LRC-columella boundary until the most distal LRC cell. To obtain a better view of the fluorescence signal in LRC cells, the ClearSee method was used[46]. Briefly, root tips from the indicated genotypes were harvested and fixed with 4% paraformaldehyde in PBS for 1 h at room temperature with gentle agitation. Fixed tissues were washed twice for 1 min in PBS and cleared with ClearSee at room temperature for 2 days. For post-staining, cleared tissues were stained with Calcofluor White (final concentration 100 μg/mL) in ClearSee solution for 1 h, and the tissues were washed in ClearSee for 1 h. For confocal imaging, 488-nm argon and 561-nm diode lasers were used for excitation.

All the images were analyzed and assembled using the image processing software ImageJ (https://imagej.nih.gov/ij) and plotted using Prism 8.0 software (GraphPad, www.graphpad.com).

## MacroView stereo microscope setup and imaging

For the time lapse imaging, the "Process Manage" function in cellSens Dimension software (Olympus) was applied for all settings[15]. In brief, an Olympus MXV10 MacroView stereo microscope was tilted by 90 degrees and adapted to a holder, which enabled imaging of the brightfield and fluorescence signals from *Arabidopsis* roots vertically growing on agar plates. A mobile microscope stage was installed to fix the plate close to the lens, and an automated ProScanTMIII system (Prior Scientific) was connected to the microscope to control the fluorescence filters. The brightfield pictures were taken every 5 min with 0.5 s exposure. For visualization of *DR5rev:3xVENUS-N7*, images were taken every 2 min with 1 s exposure depending on the expression level of the fluorescence signal. After finishing the time lapse imaging, the images were saved as video files and further analyzed by ImageJ. The fluorescence signals of DR5rev:3xVENUS-N7 were quantified in the distal salt region and the proximal salt region.

## Statolith staining and quantification

To observe the starch granules in the root tips, Lugol's staining[29] and modified propidium iodide staining (mPS-PI) staining[47] were applied. Briefly, roots were dipped in Lugol's staining solution (Sigma–Aldrich, Product No. 1.09261) for 5 min, and then immediately transferred to microscopy slides covered with chloral hydrate solution (4 g chloral hydrate, 1 mL glycerol, and 2 mL water) for observation under a dissecting microscope (Leica DM2500). For mPS-PI staining, whole seedlings were fixed in 50% methanol/10% acetic acid at 4 °C for 24 h. The seedlings were rinsed with ddH$_2$O and incubated in 1% periodic acid for 40 min. The seedlings were then rinsed twice with ddH$_2$O and incubated in Schiff's reagent containing propidium iodide (100 mM sodium metabisulfite, 0.15 N HCl, and 100 μg/mL propidium iodide) for 2 h until the seedlings were visibly stained. The seedlings were transferred onto glass slides covered with chloral hydrate solution and

kept at room temperature for one day. Subsequently, the seedlings were mounted in Hoyer's solution (30 g gum arabic, 200 g chloral hydrate, 20 g glycerol, and 50 mL H$_2$O) and left undisturbed for at least 3 days before observation under a confocal microscope (ex: 488 nm, em: 520–720 nm). The quantification of root cap amyloplasts was determined by measuring the area of the stained amyloplasts using ImageJ software[20]. The area of starch granules is obtained by raising the threshold of the image grayscale value to only retain the amyloplasts.

## Quantitative real-time PCR analysis

RNA was extracted from the root tips of the tested seedlings using TRIzol Reagent (Invitrogen). After 1 μg of total RNA was treated with RNase-free DNase (Promega), first-strand cDNA synthesis was carried out using Superscript II reverse transcriptase (Invitrogen) according to the manufacturer's instructions. The levels of the target genes were normalized to the levels of two independent reference genes, *EF-1α* (AT1G07920) and *EXPRS* (AT2G32170). qRT–PCR analysis was performed on a QuantStudio 6 Flex Real-Time PCR System apparatus with the dye SYBR Green (Invitrogen). All the individual reactions were performed in triplicate. The primers used for qRT–PCR analysis are listed in Supplementary Table 1.

## Yeast one-hybrid assay

For the yeast one-hybrid (Y1H) assay, the full-length coding sequence of *SMB* was cloned and inserted into the pGADT7 (AD) vector to construct fusion proteins containing the yeast GAL4 transcription activation domain. The F1/F2 fragment from the *AUX1* promoter region from −2710 to −2197 bp and from −1970 to −843 bp was amplified (Supplementary Table 1) and cloned and inserted into the pAbAi vector. Then, pGADT7 carrying SMB and pAbAi carrying different sequences were co-transformed into the Y1H Gold yeast strain. AD-p53 was transformed into Y1HGold (p53-AbAi) as a positive control. A further assay was performed using the Matchmaker Gold yeast One-Hybrid Library Screening System (Clontech, Takara). The minimal growth-inhibitory concentration of aureobasidin A for the Bait-Reporter yeast strain was determined.

## ChIP assay

1.0 g of roots from *p35S:GFP* or *pSMB:SMB-GFP/smb-3* plants were harvested for chromatin immunoprecipitation (ChIP) assay[48]. The samples were fixed with 1% formaldehyde, and the nuclei were extracted and sheared by sonication (Bioruptor Pico, Diagenode) to obtain chromatin DNA fragments with an average size of 500 bp. Anti-GFP antibodies (Abmart, M20004) preincubated with protein A agarose beads were used to immunoprecipitate genomic DNA fragments. PCR was performed with the immunoprecipitated genomic DNA fragments. Primers were designed across the SMB binding site of the *AUX1* promoter and near the translation initiation codon (ATG) as a negative control (F3/R3). The primer sequences used are listed in Supplementary Table 1. The band intensities were quantified by using ImageJ software, and the enrichment of SMB relative to the input was calculated.

## Data analyses

The experiments performed in this study were repeated at least three times, and all the results are presented as the mean ± SD. A significant difference between two sets of data was determined by Student's *t* test, whereas differences among more than two sets of data were analyzed by one-way ANOVA or two-way ANOVA with post-hoc tests (Tukey's multiple comparisons test) using GraphPad Prism 8 software.

## Reporting summary

Further information on research design is available in the Nature Portfolio Reporting Summary linked to this article.

## Data availability

All relevant data supporting the findings of this study are provided in the main figures and Supplementary Information files, and are available from the corresponding author upon request. Source data are provided with this paper.

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

## Acknowledgements

We thank Yu Wang and Tianru Zhang for the help of mutant screening, Prof. Guanghui Xiao (Shaanxi Normal University) for providing *ss4-3* and *adg1-1* mutants, Hongye Qu and Xiaoli Dai (Nanjing Agricultural University) for their technical help with the confocal imaging, and Dr. Hugues De Gernier (Ghent University) for helpful comments on the manuscript. This work was supported by China National Key Program for Research and Development (2021YFF1000403), the Project of Sanya Yazhou Bay Science and Technology City (SCKJ-JYRC-2022-21), the National Natural Science Foundation (32072658), the Natural Science Foundation of Anhui Province (2208085MC44), the start-up funding of Anhui Agricultural University, and the Fundamental Research Funds for the Central Universities (KYT2023001 and XUEKEN2023043). L.Z. is supported by grant from the Chinese Scholarship Council (CSC, 202206850039).

## Author contributions

Y. Han., W.X. and T.B. supervised and designed the study. L.Z. performed most of the experiments. Y. Hu. helped the ChIP assay. T.Y., Z.W. and D.W. generated the genetic materials. L.J. helped the Na+ influx assay. Y.X. and L.L. helped with imaging. W.Q. and Y.L. helped with data analysis. All authors discussed the results and contributed to the finalization of the manuscript.

## Competing interests

The authors declare no competing interests.
