## [Peer Review File · Nature Communications]

A root cap-localized NAC transcription factor controls root halotropic response to salt stress in ArabidopsisReviewer #1 (Remarks to the Author):

The authors have done an elaborate mutant screen to find new players in regulation of halotropism. The *smb-3* mutant identified shows a clear reduction in negative root halotropism and the thorough follow-up studies align well with the existing literature and provide major new insight. This manuscript is a major significant advance on the regulation of the NaCl-induced asymmetrical auxin distribution since the original publication of halotropism in 2013. The main conclusion that spatiotemporal regulation of AUX1 in the root cap by SMB is a central factor in halotropism is well supported by the data. Yet, while the results generally align very well with the conclusions, I have some major concerns that need to be addressed and several minor suggestions to improve the manuscript.

1. Most importantly, the description on the quantification of the root angle in materials and methods is unclear. Namely, the authors state that "The bending angles of the roots away from the vertical direction (0°) were measured and analyzed using ImageJ software. (350-351)". Does this imply that root angle was measured (1) from the moment that the gradient was created, (2) the whole root or (3) only the apical root?. In Figure S1C for example, it can be observed that 35S:SMB-GR roots at 1 μM DEX and 150 mM NaCl grow toward the salt (in fact they look agravitropic rather than more sensitive to the gradient) but the quantification in S1d only shows angles away from the salt.

2. Furthermore, the manuscript would greatly benefit from quantification of root growth and an indication of the location of the root tip at the moment of the introduction of the gradient. Namely, from the current images, it seems that some roots (Figure 3E,G and the *smb-3* roots of Figure 4A and S1A) are barely growing. In Figure 1 after 3 days this is less apparent, but after 1 day there seems to be a delay in growth. I assume growth rates can be calculated from the current data as the seedlings have been placed at the same position at the start (0.5 cm from the cut) and were imaged at multiple days. Sometimes data of 1 day and sometimes 3 days after introducing the gradient are shown, it would anyways be good to have all growth and angle data in supplemental info. This is important as strongly reduced root growth could look like a strong inhibition of bending, while it actually shows that roots do not have enough cell elongation to facilitate root bending.

3. The amyloplast data in Fig S5 and Fig 2A would need to be quantified. Also, the two stains (Lugol's and mPS-PI) appear to give contradicting results, but this is not adequately discussed. From FigS5 it seems that amyloplast degradation in salt is slower in the *smb-3* mutant. It is reported that "To our surprise, the mPS-PI staining showed that no decreased amount of amyloplasts was observed in both Col-0 and *smb-3* columella after halo-stimulation in comparison with the corresponding control treatments (Fig. 2A)." I am puzzled by this statement. Is the difference caused by solid media vs gradient assays (and thus actual salt concentration experienced by the root cap cells), then why different stains? As the concentration NaCl during halosimulation is likely $<150\text{mM}$ at the root tip, the data could in fact be consistent. Please provide quantified data for both stains in both setups to resolve this point.

Minor comments:

1-introduction

line 49 "... contributes to enhancing stress resilience", this statement sounds logical but as far as I know is not supported by literature, please revise.

Line 72. For hydrotropism it has been shown by Dietrich et al 2017

(<http://dx.doi.org/10.1038/nplants.2017.57>) that columella cells are dispensable for the hydrotropic response, so consider deleting hydrotropism from this sentence. Also, the role of auxin in hydrotropism is quite unclear still.

2. Results and discussion

Line 101 "displayed a compelling halotropic-avoiding phenotype" would be clearer to rephrase to indicate the phenotype is a lack of halotropic response; the mutant does not avoid salt in a

gradient assay.

Line 151 rephrase for grammar

Line 227 the finding is also in line with the effect of addition of 10nM IAA as previously reported in ref 5.

Line 259 the differential AUX1 abundance at salt-exposed and non-exposed side was also previously shown in ref 8 for wt Col-0. Results shown here are consistent with those data.

Line 279 again would be relevant to mention that results are in accordance with the conclusions of that same paper (and ref 7) where mathematical modelling and experimental data already supported the essential role of AUX1 in the halotropic response.

3. Materials and methods

Line 342 reference is 5, not 4

4. Fig. 4A/B It is quite hard to see the difference in DR5 signal in the images. Are the roots stained? Which region is quantified exactly in B? Please provide this information in the legend.

Other general minor comment

It would be interesting to include the results of the screening of the 6866 T-DNA insertion lines mentioned. It would be useful for the community to know which other lines potentially have a halotropism phenotype and which not.

Reviewer #2 (Remarks to the Author):

The authors have presented a study detailing the role of the NAC transcription factor SMB in regulating halotropic responses in Arabidopsis roots. The authors have done a very good job of dissecting and analyzing a component of a very complex signaling pathway consisting of multiple overlapping and integrated signals (auxin, statoliths, etc.). The experiments are generally well-designed and the manuscripts follows a logical progression. The supplementary data are clearly presented and support the hypotheses put forward in the main body of the manuscript. The main figures are concise and clear and the conclusions drawn are supported by the data presented. The data presented significantly increases knowledge in the field.

Minor points:

-The manuscript overall would benefit from an additional round of editing with a focus on grammar and verb tense.

-In several places, the meaning of the authors is a bit confusing or unclear and this text should be revised to increase clarity. Examples where this include:

--Lines 143-148 discussing the intersection between gravitropism and halotropism.

--Lines 159-160

--Lines 168-169 and 174-175 also discussing the intersection between gravitropism and halotropism.

-In lines 255-256, the text should be revised more clearly summarize why the data supports a positive role for SMB in regulating AUX1 transcription.

-It would be helpful if the model presented in Fig. S13 could be incorporated into the primary figures for the manuscript. This would allow readers to better place the data presented in the manuscript into current models of root halotropism.

We thank both reviewers very much for the constructive comments and suggestions on improving our manuscript. We have carefully considered the comments and suggestions and revised the manuscript to address them. Please find our point-by-point responses below. The paragraphs in blue were our responses. Revised contents in the manuscript were marked in blue.

Reviewer #1 (Remarks to the Author): The authors have done an elaborate mutant screen to find new players in regulation of halotropism. The *smb-3* mutant identified shows a clear reduction in negative root halotropism and the thorough follow-up studies align well with the existing literature and provide major new insight. This manuscript is a major significant advance on the regulation of the NaCl-induced asymmetrical auxin distribution since the original publication of halotropism in 2013. The main conclusion that spatiotemporal regulation of AUX1 in the root cap by SMB is a central factor in halotropism is well supported by the data. Yet, while the results generally align very well with the conclusions, I have some major concerns that need to be addressed and several minor suggestions to improve the manuscript.

1. Most importantly, the description on the quantification of the root angle in materials and methods is unclear. Namely, the authors state that “The bending angles of the roots away from the vertical direction (0°) were measured and analyzed using ImageJ software. (350-351)”. Does this imply that root angle was measured (1) from the moment that the gradient was created, (2) the whole root or (3) only the apical root?. In Figure S1C for example, it can be observed that 35S:SMB-GR roots at 1 µM DEX and 150 mM NaCl grow toward the salt (in fact they look agravitropic rather than more sensitive to the gradient) but the quantification in S1d only shows angles away from the salt.

Answer: We agree with the reviewer’s comment. We have re-written the description on the quantification of root bending angle as below paragraph. Accordingly, we also supplemented a schematic diagram for the assistance in understanding (See below figure as in new Supplementary Fig. 1).

“The capacity of the roots to grow away from the salt gradient was conceptualized as a bending angle as shown in Supplementary Fig. 1. For each root, the bending angle was quantified by first drawing a straight line connecting the locations of the root apex upon halo-stimulation (i.e., the start/zero time point) and after the indicated number of days. Second, the angle was measured between the drawn line and the gravity vector. The bending angle was assessed after one and three days of growth. All the data were measured and analysed using ImageJ software.” (See lines 395–401 in the revised manuscript text).

Supplementary Figure 1. Quantification of root halotropic curvature. Related to Fig 1.

a, b Representative images of root halotropic growth for 1 day (**a**) and 3 days (**b**) following halo-stimulation. A straight line was drawn connecting the location of the root tip at the indicated time points and at the zero time point during halo-stimulation, and the angle of the root bending from the vertical (set to 0°) was measured. α_1 and α_3 represent the bending angles at 1 day (**a**) and 3 days (**b**) of halo-stimulation, respectively. **c** Diagram showing the angles of the root tip response to salinity at the indicated time points following halo-stimulation. Percentages of roots in angle categories of 10 degrees, with the number (*n*) of roots measured, were quantified. Vertical angle sets to 0°. The angle away or toward from the salt gradient is defined as a positive (+) or negative value (-), respectively. The length of the bars represents the relative number of roots per category.

2. Furthermore, the manuscript would greatly benefit from quantification of root growth and an indication of the location of the root tip at the moment of the introduction of the gradient. Namely, from the current images, it seems that some roots (Figure 3E,G and the smb-3 roots of Figure 4A and S1A) are barely growing. In Figure 1 after 3 days this is less apparent, but after 1 day there seems to be a delay in growth. I assume growth rates can be calculated from the current data as the seedlings have been placed at the same position at the start (0.5 cm from the cut) and were imaged at multiple days. Sometimes data of 1 day and sometimes 3 days after introducing the gradient are shown, it would anyways be good to have all growth and angle data in supplemental info. This is important as strongly reduced root growth could look like a strong inhibition of bending, while it actually shows that roots do not have enough cell elongation to facilitate root bending.

Answer: We much appreciate this nice advice. Yes, as the reviewer suggested, in the revised manuscript, primary root elongation was calculated by measuring the length of newly grown root at the indicated time points since halo-stimulation, and new graphs were made and incorporated into the figures including new Supplementary Fig. 2e, new Supplementary Fig. 3b, new Supplementary Fig. 4c, new Supplementary Fig. 5c, new Supplementary Fig. 8e, new Supplementary Fig. 11c, new Supplementary Fig. 12c, new Supplementary Fig. 13e, new Supplementary Fig. 15e, new Supplementary Fig. 17c, d, i, new Supplementary Fig. 19c, new Supplementary Fig. 20c, and new Supplementary Fig. 21c, e. The location of the root tip at the moment when the salt gradient was established

has also been marked with a horizontal dotted line that occurred in the related figures.

The angle data of halotropic bending including phenotype and quantification at 1 and 3 days of halo-stimulation were provided in the revised manuscript. Apart from already having angle data in the original version, newly added figures include new Fig. 3e–h, new Supplementary Fig. 2c, d, new Supplementary Fig. 3a, new Supplementary Fig. 5a, b, new Supplementary Fig. 11a, b, new Supplementary Fig. 13a, b, new Supplementary Fig. 17a, b, g, h, new Supplementary Fig. 19b, and new Supplementary Fig. 21d.

Thanks to the suggestion from the reviewer on the quantifying the root growth and indicating the location of the root tip at the moment when the gradient was created, new evidence by measuring root elongation support the conclusion that the loss of halotropic bending of *smb-3* does not result from defects in root elongation. This is evident from the observations that (1) quantification of the root elongation in *smb-3* at 3 days after exposure to a 250 mM NaCl gradient clearly showed that *smb-3* root remained elongating and could enter the salt-containing area while Col-0 root was observed to grow away from salt-containing area (new Supplementary Fig. 2c-e; new Supplementary Fig. 3b); (2) halotropic root responses of auxin antagonists-treated *smb-3* were also restored independently of primary root elongation inhibition (new Supplementary Fig. 15e); (3) the *aux1-21 smb-3* double mutant showed enhanced root bending phenotype in response to salt gradients (Fig. 4a), while its root elongation was similar with that of *smb-3* (new Supplementary Figure 19c). These results suggest that root elongation defects could not be linked with altering halotropic root growth of *smb-3*. We have further added the related description and discussion in the revised manuscript (lines 104–110, lines 221–224).

3. The amyloplast data in Fig S5 and Fig 2A would need to be quantified. Also, the two stains (Lugol's and mPS-PI) appear to give contradicting results, but this is not adequately discussed. From FigS5 it seems that amyloplast degradation in salt is slower in the *smb-3* mutant. It is reported that "To our surprise, the mPS-PI staining showed that no decreased amount of amyloplasts was observed in both Col-0 and *smb-3* columella after halo-stimulation in comparison with the corresponding control treatments (Fig. 2A)." I am puzzled by this statement. Is the difference caused by solid media vs gradient assays (and thus actual salt concentration experienced by the root cap cells), then why different stains? As the concentration NaCl during halosimulation is likely <150mM at the root tip, the data could in fact be consistent. Please provide quantified data for both stains in both setups to resolve this point.

Answer: Many thanks for this crucial suggestion. Indeed, the amyloplast data from Fig S5 and Fig 2A seem to be inconsistent at a glance. However, as mentioned by the reviewer, this is possibly due to that the actual salt concentration sensed by the root cap cells under solid media versus gradient assays could be different. In specific, using the split-agar medium system, root tips experience a gradual increase of NaCl, raising up to about 15 % and 26 % of the initial concentration of the salt-containing medium at 12 hrs and 24 hrs of halo-stimulation respectively, as reported in Galvan-Ampudia et al. (2013). When the

seedlings were placed at the maximal concentration of salt gradient (at 250 mM) tested in this work, the estimated NaCl concentration experienced by the root cap cells would reach about 37.5 mM and 65 mM at 12 hrs and 24 hrs since halo-stimulation respectively. In Fig 2A, the sampling time for amyloplast staining was 6 hrs following halo-stimulation. It could be expected that the actual salt concentration experienced by the root cap cells should be less than 37.5 mM. In line with the previous findings (Sun et al., 2008), our new data showed that there was no apparent reduction of amyloplasts in Col-0 and *smb-3* under low NaCl (below 75 mM)-containing solid media during 24 hrs of treatment; the rapid amyloplast degradation could be triggered as the actual concentrations of salt at the root tip reach at 100 mM and beyond (new Supplementary Fig. 9). Therefore, it could be acceptable that no decreased amount of amyloplasts was observed in both Col-0 and *smb-3* columella cells of roots during the initial 24 hrs of halo-stimulation at 250 mM and lower NaCl gradients (Fig. 2a–c; new Supplementary Fig. 10).

To clarify this issue, we further provided additional data including (1) the two stains (Lugol's and mPS-PI) and their quantification analysis for Col-0, *smb-3*, *ss4-3*, *ss4-3 smb-3*, *adg1-1* and *adg1-1 smb-3* (new Fig. 2a, b); (2) the two stains and their quantification analysis for Col-0 and *smb-3* over 24 hrs grown under a split-agar/gradient medium containing 150 or 250 mM NaCl (new Supplementary Fig.10); (3) the two stains and their quantification analysis for Col-0 and *smb-3* over 24 hrs grown under full NaCl (50 to 250 mM)-containing solid media (new Supplementary Fig. 9). Both of two stains showed identical results. Accordingly, the related result description and discussion were added into the revised manuscript (lines 168–183). In addition, the original Fig S5 and Fig 2A have been removed and replaced by above-mentioned figures.

References

Galvan-Ampudia CS, Julkowska MM, Darwish E, Gandullo J, Korver RA, Brunoud G, Haring MA, Munnik T, Vernoux T, Testerink C. Halotropism is a response of plant roots to avoid a saline environment. *Curr Biol*. 2013 23(20):2044–50.

Sun F, Zhang W, Hu H, Li B, Wang Y, Zhao Y, Li K, Liu M, Li X. Salt modulates gravity signaling pathway to regulate growth direction of primary roots in *Arabidopsis*. *Plant Physiol*. 2008 146(1):178–88

Minor comments:

1-introduction

line 49 “.. contributes to enhancing stress resilience”, this statement sounds logical but as far as I know is not supported by literature, please revise.

Answer: Yes, it has been removed and revised to “...likely contributes to plant acclimatory responses to environmental stresses” (lines 43–44).

Line 72. For hydrotropism it has been shown by Dietrich et al 2017

(<http://dx.doi.org/10.1038/nplants.2017.57>) that columella cells are dispensable for the hydrotropic response, so consider deleting hydrotropism from this sentence. Also, the role of auxin in hydrotropism is quite unclear still.

Answer: Deleted.

2. Results and discussion

Line 101 “displayed a compelling halotropic-avoiding phenotype” would be clearer to rephrase to indicate the phenotype is a lack of halotropic response; the mutant does not avoid salt in a gradient assay.

Answer: Agreed. It has been modified as “was found to be almost incapable of bending away from such a salt gradient” (lines 98–99).

Line 151 rephrase for grammar

Answer: Rephrased (line 162).

Line 227 the finding is also in line with the effect of addition of 10nM IAA as previously reported in ref 5.

Answer: Yes, the related information including reference was integrated into the text (line 261).

Line 259 the differential AUX1 abundance at salt-exposed and non-exposed side was also previously shown in ref 8 for wt Col-0. Results shown here are consistent with those data.

Answer: We agree with that. The related information was added (lines 295–297).

Line 279 again would be relevant to mention that results are in accordance with the conclusions of that same paper (and ref 7) where mathematical modelling and experimental data already supported the essential role of AUX1 in the halotropic response.

Answer: We agree and have added the related content and cited the references (lines 321–322).

3. Materials and methods

Line 342 reference is 5, not 4

Answer: Corrected.

4. Fig. 4A/B It is quite hard to see the difference in DR5 signal in the images. Are the roots stained? Which region is quantified exactly in B? Please provide this information in the legend.

Answer: Thanks very much for this comment. The Fig.4A represented the images shown in the overlay of VENUS and PI channels. Now, the images from Fig. 3A and 4A were replaced to those only shown in the VENUS channel in the new Fig. 3a and 4a, as the images shown in PI channel and the overlay of both VENUS and PI were deposited in the new supplementary Fig. 16 and new supplementary Fig. 19a. Also, the information about quantification of DR5 signal in new Fig. 3b and 4b was provided in the legend, as follows.

“The fluorescence intensity of DR5rev:3xVENUS-N7 in the LRC and epidermis distal (left) or proximal (right) to mock/salt gradient was quantified.”

Schematic diagram of fluorescence measurement.

Fluorescence intensity of DR5rev:3xVENUS-N7 signal is measured in white dotted boxes area which represent the LRC and epidermis cells. The root area away from the mock/salt gradient was named the distal (Left), and the corresponding side was named the proximal (Right). White arrows indicate the direction of NaCl diffusion, while dashed white lines denote the tentative outline of the root tip. Scale bar, 100 μ m.

Other general minor comment

It would be interesting to include the results of the screening of the 6866 T-DNA insertion lines mentioned. It would be useful for the community to know which other lines potentially have a halotropism phenotype and which not.

Answer: We thank the reviewer for this advice. Besides *smb-3*, we also identified *phyB* (SALK_022035C) and *phyC* T-DNA lines (SALK_057517C; Below figure) that showed defect halotropic root response under halo-stimulation of 250 mM salt gradient. We are quite interested in understanding how these light photoreceptors affect root halotropism and the underlying mechanism. This project is currently in progress. Hopefully, we could give more exhaustive and/or clearer information potentially connecting light signalling and root halotropism after we have studied their effects in details.

(a and b) Halotropic root response of Col-0, *smb-3*, *phyB* and *phyC* seedlings that were transferred to split-agar medium containing 250-mM NaCl for 3 days **(a)**. Orange and red arrows represent the presence and absence of halotropic root bending, respectively. Scale bar, 1 cm. Halotropic root curvature was quantified and shown in **(b)**. Violin plot showing the distribution ($n \geq 10$ seedlings). Alphabets denote significant differences ($P < 0.05$, one-way ANOVA by Tukey's test).

Reviewer #2 (Remarks to the Author):

The authors have presented a study detailing the role of the NAC transcription factor SMB in regulating halotropic responses in Arabidopsis roots. The authors have done a very good job of dissecting and analyzing a component of a very complex signaling pathway consisting of multiple overlapping and integrated signals (auxin, statoliths, etc.). The experiments are generally well-designed and the manuscripts follows a logical progression. The supplementary data are clearly presented and support the hypotheses put forward in the main body of the manuscript. The main figures are concise and clear and the conclusions drawn are supported by the data presented. The data presented significantly increases knowledge in the field.

Minor points:

-The manuscript overall would benefit from an additional round of editing with a focus on grammar and verb tense.

Answer: Many thanks for the suggestion. We have re-examined the English throughout the text, based on both reviewers' advice. Moreover, grammar, spelling, verb tense and other common errors in the manuscript have been checked through a language editing service provided through the website of Nature Communications. A certificate of language editing has been uploaded.

-In several places, the meaning of the authors is a bit confusing or unclear and this text should be revised to increase clarity. Examples where this include:

--Lines 143-148 discussing the intersection between gravitropism and halotropism.

Answer: Yes, we have added the discussion on intersection between gravitropism and halotropism (lines 156–159), as follows.

“These results indicate that SMB is not required for gravitropism but plays a specific role in root halotropism. Based on these findings, we propose that SMB may act downstream of salt stress signalling and that salt-activated root halotropism occurs prior to gravitropism.”

--Lines 159-160

Answer: It has been revised as follows: “Altogether, these results imply that halotropic root responses mediated by SMB are less closely linked to alterations in amyloplast sedimentation.” (lines 182–183).

--Lines 168-169 and 174-175 also discussing the intersection between gravitropism and halotropism.

Answer: Yes, we have added the discussion on intersection between gravitropism and halotropism as follows:

“These two opposite findings may hint that gravitropism antagonizes halotropic root bending. The enhanced halotropic root bending in *ss4-3* and *adg1-1* is therefore likely to be caused by impaired gravitropism, but not enhanced halotropism.” (lines 191–194).

“Again, amyloplast-dependent gravity signalling pathways appear not to be involved in SMB-mediated root halotropism.” (lines 200–201).

-In lines 255-256, the text should be revised more clearly summarize why the data supports a positive role for SMB in regulating AUX1 transcription.

Answer: We are sorry for our inaccurate expression. Now it has been modified as follows. “Collectively, these results demonstrated that SMB could bind to the promoter of AUX1 and maintain its basal expression.” (lines 293–294).

-It would be helpful if the model presented in Fig. S13 could be incorporated into the primary figures for the manuscript. This would allow readers to better place the data presented in the manuscript into current models of root halotropism.

Answer: Good suggestion! The Fig. S13 in the previous version of manuscript was moved into the main figures as shown in new Fig. 5 in the revised manuscript.

Reviewer #1 (Remarks to the Author):

The authors have adequately addressed all previous comments in our opinion, congratulations on this work.

Reviewer #2 (Remarks to the Author):

The authors have presented a substantially revised manuscript, which substantially addresses the points raised in the previous review. The discussion detailing the differences and interactions between halotropism and gravitropism is much improved. The revised figures also are also very helpful and informative.